# Framework for rapid comparison of extracellular vesicle isolation methods

Dmitry Ter-Ovanesyan[1†], Maia Norman[1,2,3†], Roey Lazarovits[1], Wendy Trieu[1], Ju-Hyun Lee[1], George M Church[1,4], David R Walt[1,3,4]*

[1]Wyss Institute for Biologically Inspired Engineering, Boston, United States; [2]Tufts University School of Medicine, Boston, United States; [3]Department of Pathology, Brigham and Women's Hospital, Boston, United States; [4]Harvard Medical School, Boston, United States

**Abstract** Extracellular vesicles (EVs) are released by all cells into biofluids and hold great promise as reservoirs of disease biomarkers. One of the main challenges in studying EVs is a lack of methods to quantify EVs that are sensitive enough and can differentiate EVs from similarly sized lipoproteins and protein aggregates. We demonstrate the use of ultrasensitive, single-molecule array (Simoa) assays for the quantification of EVs using three widely expressed transmembrane proteins: the tetraspanins CD9, CD63, and CD81. Using Simoa to measure these three EV markers, as well as albumin to measure protein contamination, we were able to compare the relative efficiency and purity of several commonly used EV isolation methods in plasma and cerebrospinal fluid (CSF): ultracentrifugation, precipitation, and size exclusion chromatography (SEC). We further used these assays, all on one platform, to improve SEC isolation from plasma and CSF. Our results highlight the utility of quantifying EV proteins using Simoa and provide a rapid framework for comparing and improving EV isolation methods from biofluids.

**\*For correspondence:**
dwalt@bwh.harvard.edu

†These authors contributed equally to this work

## Introduction

Extracellular vesicles (EVs) are released by all cell types and are found in biofluids such as plasma and cerebrospinal fluid (CSF). EVs contain contents from their donor cells, providing broad non-invasive access to molecular information about cell types in the human body that are otherwise inaccessible to biopsy (*Hirshman et al., 2016*). Despite the diagnostic potential of EVs, there are several challenges that have hampered their utility as biomarkers. EVs are heterogeneous, present at low levels in clinically relevant samples, and difficult to quantify (*Tkach et al., 2018*; *Hartjes et al., 2019*; *Shao et al., 2018*). Due to these challenges, there is a lack of consensus about the best way to isolate EVs from biofluids (*Coumans et al., 2017a*; *Konoshenko et al., 2018*; *Théry et al., 2018*).

Several techniques have been used in attempts to quantify EVs. These methods, such as nanoparticle tracking analysis (NTA), dynamic light scattering, and tunable resistive pulse sensing, aim to measure both particle size and concentration (*Hartjes et al., 2019*). A major limitation of these methods is that they cannot discriminate lipoproteins or particles of aggregated proteins from EVs (*Coumans et al., 2017a*; *Sódar et al., 2016*; *Welton et al., 2015*; *Lee et al., 2019*; *Webber and Clayton, 2013*; *Johnsen et al., 2019*). In addition, they are all physical methods that provide no information about the biological nature of the particles being measured. Since biofluids, and plasma in particular, contain an abundance of lipoproteins and protein aggregates at levels higher than those of EVs (*Sódar et al., 2016*; *Simonsen, 2017*), these methods are ill-suited for quantifying EVs (*Tkach et al., 2018*). Lipid dyes have also been used to label and measure EVs (*Osteikoetxea et al., 2015*; *Visnovitz et al., 2019*), but these dyes also bind to lipoproteins and lack sensitivity (*Tkach et al., 2018*). There are also numerous efforts to apply flow cytometry to the analysis of EVs, but due to

the small size of EVs, obtaining quantitative measurements using this approach remains challenging (*Lucchetti et al., 2020*; *Kuiper et al., 2021*; *Welsh et al., 2020*).

A feature of EVs that distinguishes them from both lipoproteins and free protein aggregates is the presence of transmembrane proteins that span the phospholipid bilayer (*Simonsen, 2017*). The tetraspanins CD9, CD63, and CD81 are transmembrane proteins that are widely expressed and readily found on EVs, often referred to as 'EV markers' (*Tkach et al., 2018*). Although none of these proteins is present on every EV, measuring three tetraspanins should be a reliable proxy for EV abundance in many contexts. We reasoned that by using immunoassays to compare the levels of tetraspanins from a given biofluid, as well as albumin as a representative free protein, we could quantitatively compare the purity and yield of different EV isolation methods.

The most commonly used method for measuring proteins in biofluids is enzyme-linked immunosorbent assay (ELISA), but this technique lacks the sensitivity to detect low-abundance proteins (*Coumans et al., 2017b*). Single-molecule array (Simoa) technology, previously developed in our lab but now commercially available, converts ELISA into a digital readout (*Rissin et al., 2010*). Simoa assays can be orders of magnitude more sensitive than traditional ELISAs (*Cohen and Walt, 2019*), which is particularly useful for EV analysis as the levels of EV proteins are often low in clinical biofluid samples (*Coumans et al., 2017b*). We have previously applied Simoa to the investigation of L1CAM, a protein thought to be a marker of neuron-derived EV, showing it is not associated with EVs in plasma and CSF (*Norman et al., 2021*).

In this study, we demonstrate the application of Simoa for relative EV quantification by comparing different EV isolation methods from human biofluids. In particular, we applied Simoa to compare EV isolation methods from human plasma and CSF using three of the most commonly used isolation techniques: ultracentrifugation, precipitation, and size exclusion chromatography (SEC). By also measuring levels of albumin using Simoa, we were able to determine both relative purity and yield for each technique in the same experiment. We then applied these Simoa assays to screen several parameters of SEC and develop improved EV isolation methods from plasma and CSF, demonstrating the utility of this approach for EV analysis.

## Results
### Framework for quantifying relative EV yield and purity

We set out to quantify the relative difference in yield and purity for different EV isolation methods. Starting with aliquots of the same biofluid, we reasoned that by measuring the tetraspanins CD9, CD63, and CD81 using different isolation methods, we could directly compare EV yield. By also measuring albumin, the most abundant free protein in plasma and CSF, we could compare the purity of these methods. Using Simoa technology, an ultrasensitive digital ELISA, to measure all four of these proteins, we could compare EV yield and purity on one platform with high sensitivity (*Figure 1a*).

Although Simoa is generally used to quantify free proteins, it can also be used to analyze EV transmembrane proteins. In Simoa, unlike in traditional ELISA, individual immuno-complexes are isolated into femtoliter wells that fit only one bead per well. In a given sample, there are many more antibody-bound beads than target proteins, and therefore Poisson statistics dictate that only a single immuno-complex is present per well. This allows counting 'on wells' as individual protein molecules (*Figure 1b*). The percentage of 'on wells' can then be converted to protein concentration by comparing to a calibration curve of recombinant protein standard. We previously developed and validated Simoa assays for the proteins CD9, CD63, and CD81, showing that they are 1–3 orders of magnitude more sensitive than the corresponding standard ELISA assays with the same pairs of antibodies (*Norman et al., 2021*).

### Comparison of existing EV isolation methods

We started by using Simoa to directly compare EV isolation methods commonly used in biomarker studies. For each method, we used identical 0.5 ml samples of human plasma or CSF that were pooled and aliquoted, allowing us to directly compare the different methods. To separate EVs from cells, cell debris, and large vesicles, all samples were first centrifuged and then filtered through a 0.45-μm filter. We compared three methods and chose two variations for each method: ultracentrifugation (with or

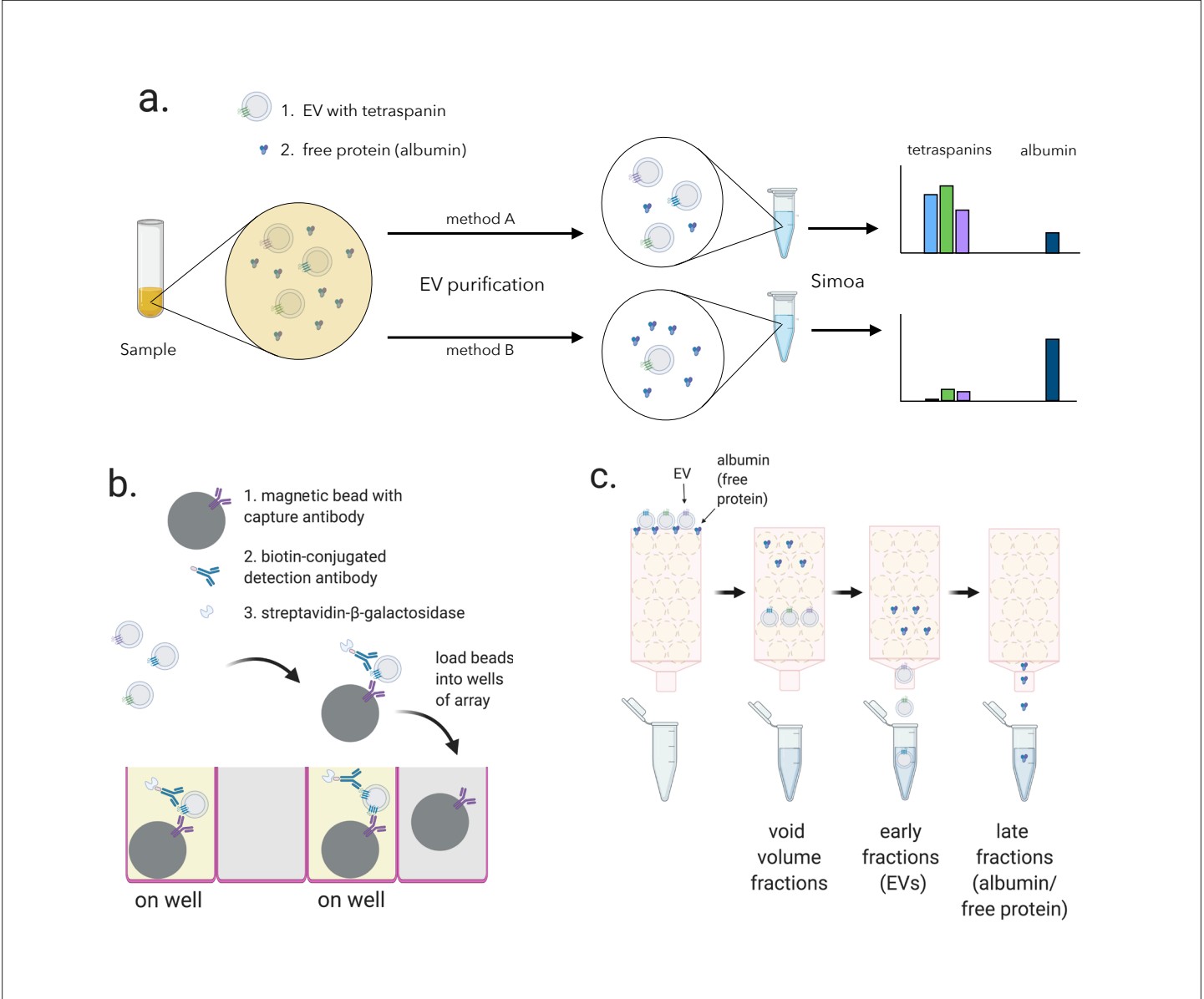

**Figure 1.** Overview of experimental framework for EV detection using Simoa and size exclusion chromatography (SEC). (**a**) Different methods of EV isolation can be directly compared to assess yield and purity by measuring the three tetraspanins (CD9, CD63, and CD81) and albumin. (**b**) Single immuno-complexes are formed by binding the target tetraspanin protein on EVs to a magnetic bead conjugated to a capture antibody and a biotin-labeled detection antibody. Detection antibodies are labeled with a streptavidin-conjugated enzyme. The beads are then loaded into individual wells of a microwell array where each well matches the size of the magnetic bead limiting a maximum of one bead per well. Wells with the full immuno-complex (on wells) produce a fluorescent signal upon conversion of substrate, unlike wells with beads lacking the immuno-complex (off wells"). (**c**) EV and free proteins such as albumin in a biofluid sample are separated by SEC. Free proteins elute from the column in later fractions than EVs because free proteins are smaller than the pore size of the beads while EVs are larger and are excluded from entering the beads. EV, extracellular vesicle.

without a wash step), two commercial precipitation kits (ExoQuick and ExoQuick ULTRA), and two commercially available SEC columns (Izon qEVoriginal 35 nm and 70 nm).

SEC separates EVs from free proteins based on size; proteins enter porous beads and elute from the column later than the EVs, which are much larger and less likely to enter the beads (*Figure 1c*). Whereas the ultracentrifugation and precipitation conditions each yielded a single sample (performed on 2 separate days and averaged; *Figure 2—figure supplement 1*), we collected several fractions for SEC and analyzed each fraction to assess the distribution of EVs relative to albumin.

We quantified EVs by measuring the levels of CD9, CD63, and CD81 across the different EV isolation methods in both plasma and CSF (*Figure 2a*). Since we are interested in all EVs, as opposed to subsets with a specific marker, we quantified EV yield by averaging the levels of the three tetraspanins. We first used the Simoa measurement (in picomoles, determined relative to a corresponding recombinant protein standard) to calculate EV recovery for each individual marker by normalizing the level of tetraspanin in each condition to the amount of that tetraspanin in fractions 7–10 of the Izon qEV 35 nm SEC column (the condition with the highest EV levels in plasma). Next, we averaged the relative tetraspanin recovery values across the three tetraspanins to calculate relative EV recovery.

After determining combined relative EV recovery and albumin concentration for each EV isolation method, we could directly compare EV recovery and purity in both plasma and CSF. In plasma, we found that the Izon qEVoriginal 35 nm SEC column (collecting fractions 7–10) yielded both the highest recovery of EVs and the highest purity (ratio of EVs to albumin) of EVs (*Figure 2b–f*). In contrast, in CSF, ExoQuick yielded the highest recovery of EVs while Izon qEVoriginal 70 nm yielded the highest purity (*Figure 2g–k*).

## Application of Simoa for custom SEC column optimization

Based on the promising results of commercial SEC columns relative to other methods, we sought to use our assays to further investigate SEC using custom columns. First, we designed an SEC stand that allows for reproducible collection of fractions and multiple columns to be run in parallel (*Figure 3— figure supplement 1*). We next took advantage of Simoa's high throughput screening capability to help identify the EV-containing fractions in SEC. This enabled us to optimize EV isolation from 0.5 ml samples of plasma and CSF using SEC. We prepared our own columns to systematically test several parameters: column height (10 or 20 ml) and resin (Sepharose CL-2B, CL-4B, or CL-6B).

This comprehensive comparison led us to several conclusions. First, we found that resins with smaller pore sizes led to higher yields of EVs. To confirm this result with another technique, we also observed the same result with Western blotting for the tetraspanins (*Figure 3—figure supplement 2*). Sepharose CL-6B, which has the smallest pore size, gave the highest yield, although it was accompanied by higher albumin contamination. For all SEC columns, higher purity could also be achieved by taking a smaller number of fractions (e.g., 7–9 instead of 7–10), albeit at the expense of lower EV yield. Additionally, we found that doubling the height of any given column from 10 to 20 ml resulted in better separation between EVs and free proteins, leading to higher purity but lower EV recovery (*Figures 3 and 4*). When we compared loading different sample volumes in a 10-ml Sepharose CL-6B column, we found that as expected, larger loading volumes led to lower purity in both plasma (*Figure 3—figure supplement 3*) and CSF (*Figure 4—figure supplement 1*).

## Direct comparison of custom SEC and previous methods

Since we used the same pools of biofluids for these experiments, combining all of the data we generated, we were able to perform a direct, quantitative comparison of the relative yields and purities of EVs across all methods tested. We analyzed these results for both plasma (*Figure 3—figure supplement 4*) and CSF (*Figure 4—figure supplement 2*). Since all Simoa measurements were performed with two technical replicates, we also confirmed that Simoa had high reproducibility between technical replicates (*Figure 3—figure supplement 5* and *Figure 4—figure supplement 1*). Our analysis shows that 10 ml Sepharose CL-6B column demonstrated the highest recovery in both plasma and CSF. The 20 ml Sepharose CL-4B column gave the highest purity (ratio of EVs to albumin) for plasma, while for CSF, the 10 ml Sepharose CL-4B column had higher purity than the 20 ml Sepharose CL-4B column. Although the 10 ml column had more albumin contamination in the EV fractions than the 20 ml column, the relative ratio of EVs to albumin was higher. Based on these results, we could select the best custom SEC column for either high yield or high purity isolation (*Table 1*).

## Comparison of top custom SEC methods for plasma and CSF

Based on our results surveying the different SEC resins and column heights, we performed additional experiments to more accurately quantify the best high yield and high purity SEC methods for plasma and CSF using another batch of biofluids with more replicates (four columns per condition). For both plasma and CSF, we compared the Sepharose CL-2B 10 ml column, used in the original SEC EV isolation publication (*Böing et al., 2014*) and in most subsequent SEC publications (*Monguió-Tortajada*

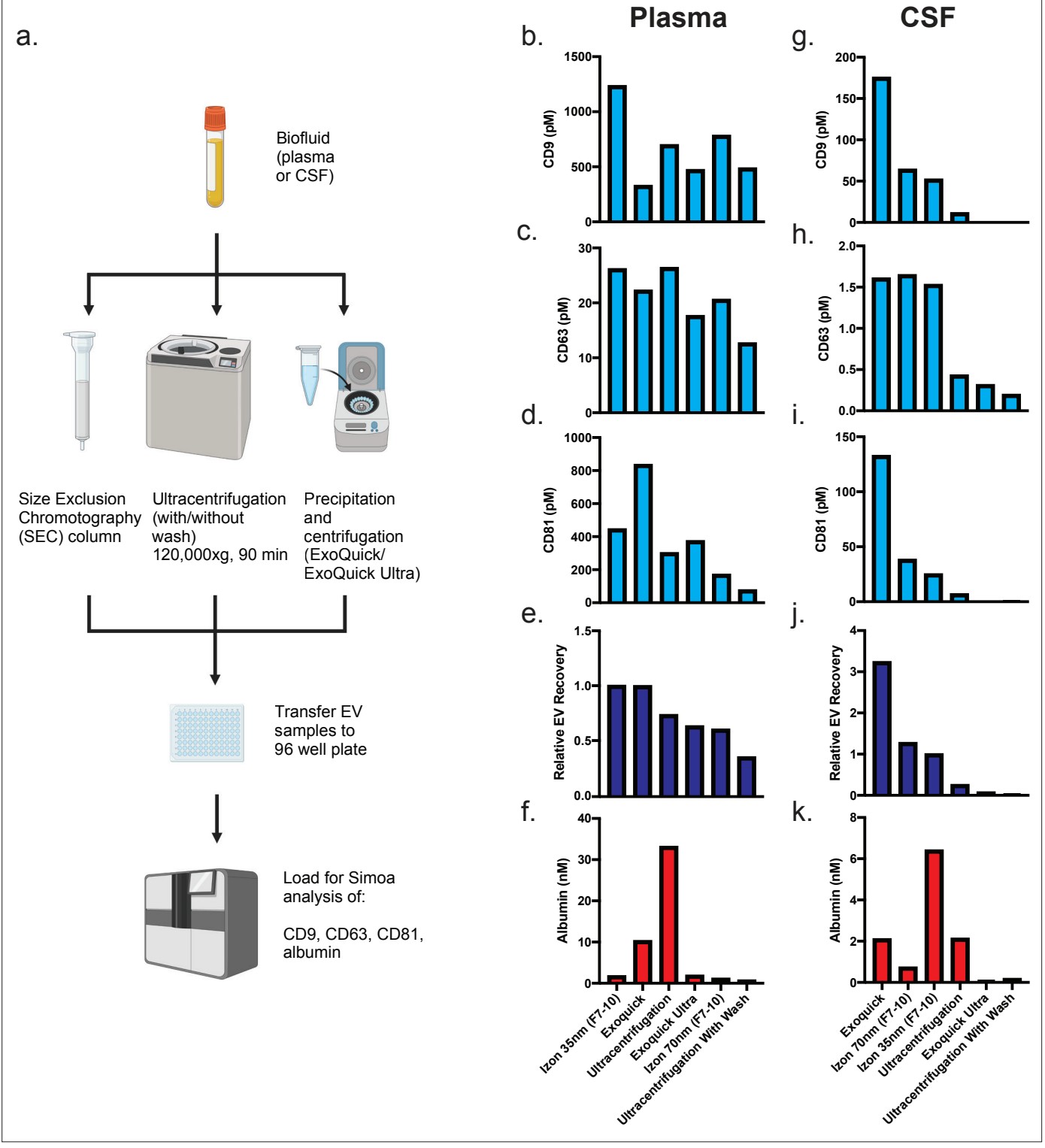

**Figure 2.** Comparison of existing methods for EV isolation in plasma and CSF. (**a**) Schematic of experimental outline. (**b–d**) Individual tetraspanin yields using different isolation methods from plasma. (**e**) Relative EV recoveries from plasma were calculated by first normalizing individual tetraspanin values (in pM) in each technique to those of Izon qEVoriginal 35 nm EV fractions 7-10 and then averaging the three tetraspanin ratios. (**f**) Albumin levels using different EV isolation methods from plasma. (**g–i**) Individual tetraspanin yields using different isolation methods from CSF. (**j**) Relative EV recoveries in CSF were calculated by first normalizing individual tetraspanin values (in pM) in each technique to those of Izon qEVoriginal 35 nm fractions 7-10 and then averaging the three tetraspanin ratios. (**k**) Albumin levels using different EV isolation methods from CSF. CSF, cerebrospinal fluid; EV, extracellular

*Figure 2 continued on next page*

*Figure 2 continued*

vesicle.

The online version of this article includes the following source data and figure supplement(s) for figure 2:

**Source data 1.** Comparison of existing methods for EV isolation in plasma and CSF.

**Figure supplement 1.** Assay reproducibility between Simoa measurements of EV isolations on 2 different days.

*et al., 2019*), to the 'high yield' Sepharose CL-6B 10 ml column. We also included a Sepharose CL-4B column as the 'high purity' column but, as plasma has much higher protein concentration than CSF, used 20 ml of resin for plasma and 10 ml for CSF.

Our results allow us to directly quantify the difference in EVs and albumin across these methods (*Figure 5*). We found that, in plasma, the Sepharose CL-6B 10 ml column provided over twofold more EVs relative to the Sepharose CL-2B 10 ml column, but also sixfold more albumin. The Sepharose CL-4B 20 ml column, on the other hand, had similar EV levels to that of Sepharose CL-2B 10 ml column in plasma but had sixfold less albumin (*Figure 5a–e*), demonstrating a large increase in relative purity (EV to albumin ratio) (*Figure 5f*). In CSF, the Sepharose CL-6B 10 ml column led to a large increase in EV yield relative to the Sepharose CL-2B 10 ml column (*Figure 5g–k*), but the Sepharose CL-4B 10 ml column did not lead to improved purity (*Figure 5l*).

## Discussion

In this study, we describe a framework for rapidly quantifying relative EV yield and purity across isolation methods, overcoming the limitations of other commonly used methods used for EV analysis. Several techniques, such as NTA and other methods developed for analysis of synthetic particles, have been applied to EV detection (*Hartjes et al., 2019*). The utility of these techniques is hindered, however, by an inability to differentiate heterogeneous EVs from other particles with overlapping size, such as lipoproteins or aggregated protein particles. Thus, although previous reports comparing EV isolation methods *Lobb et al., 2015*; *Helwa et al., 2017*; *Baranyai et al., 2015*; *Soares Martins et al., 2018*; *An et al., 2018*; *Stranska et al., 2018*; *Diaz et al., 2018*; *Kalra et al., 2013*; *Serrano-Pertierra et al., 2019*; *Gámez-Valero et al., 2016*; *Takov et al., 2019*; *Brennan et al., 2020* have yielded some useful insights, the lack of reliable EV quantification has made these studies difficult to interpret (*Tkach et al., 2018*; *Coumans et al., 2017a*; *Ludwig et al., 2019*).

The measurement of EV transmembrane proteins overcomes the limitations of EV quantification with particle detection methods. Since the transmembrane proteins CD9, CD63, and CD81 are present on EVs but are not present in lipoproteins or free protein aggregates, these tetraspanins can be used for relative quantification of EVs. Although not every EV necessarily contains a tetraspanin protein, by detecting three different tetraspanins per sample with Simoa, we minimize the chance that we are measuring a rare subset of EVs. In the experiments reported here, we observed a strong correlation of the relative levels of the three tetraspanins in different SEC fractions. Since we compared isolation methods from the same starting sample, we were able to provide a direct quantitative comparison of tetraspanin levels between the different isolation methods.

We used Simoa in this study, which is particularly well suited for EV analysis due to the technology's high dynamic range, throughput, and sensitivity. This sensitivity is achieved by converting ELISA to a digital readout via immuno-capture and counting of individual protein molecules in a microwell array. We used the commercially available Quanterix HD-X instrument, but our lab has also developed other digital ELISA methods using commonly available instrumentation (*Cohen et al., 2020*; *Maley et al., 2020*; *Wu et al., 2020*), which could be similarly applied to EVs. One could also follow a similar approach to the one we present here with traditional ELISA or other protein detection methods, but we find that high sensitivity is often necessary for the low levels of EVs in human biofluids. We have previously shown in a direct comparison (using the same antibodies) that Simoa can detect EV markers in cases where traditional ELISA cannot, such as SEC fractions of CSF (*Norman et al., 2021*).

We used Simoa to directly compare the yield and purity of commonly used EV isolation methods. To obtain the purest EVs possible (and separate EVs from lipoproteins), it has been demonstrated that

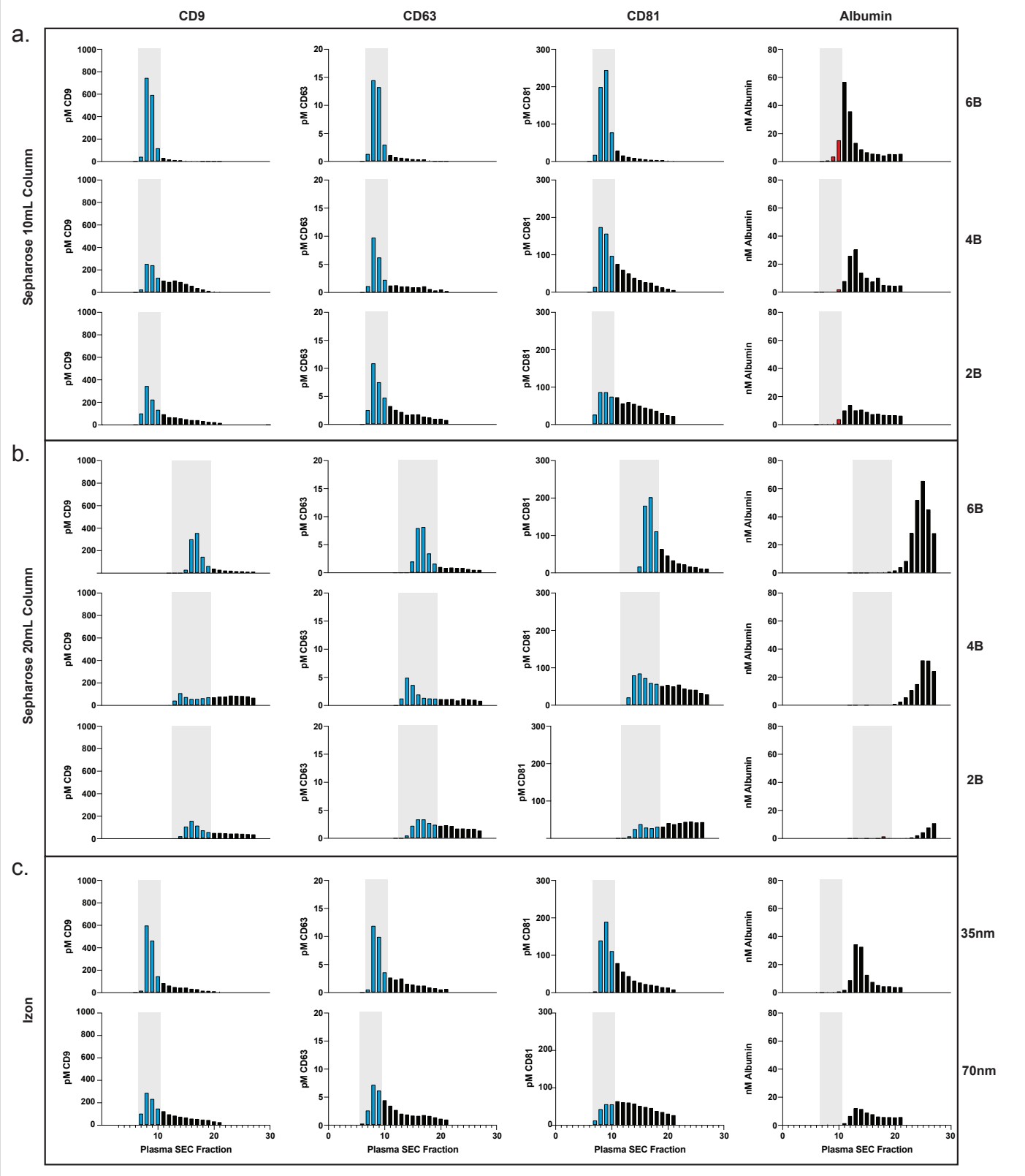

**Figure 3.** Comparison of SEC methods for EV isolation in plasma. (**a**) Levels of tetraspanins and albumin in plasma after fractionation with 10 ml custom columns filled with Sepharose CL-6B (top), Sepharose CL-4B (middle), and Sepharose CL-2B (bottom). (**b**) Levels of tetraspanins and albumin in plasma after fractionation with Izon qEVoriginal 35 nm column (top) and Izon qEVoriginal 70 nm column (bottom). (**c**) Levels of tetraspanins and albumin in plasma after fractionation with 20 ml custom columns; Sepharose CL-6B (top), Sepharose CL-4B (middle), and Sepharose CL-2B (bottom). EV,

*Figure 3 continued on next page*

*Figure 3 continued*

extracellular vesicle; SEC, size exclusion chromatography.

The online version of this article includes the following source data and figure supplement(s) for figure 3:

**Source data 1.** Plasma SEC optimization.

**Figure supplement 1.** Custom stand designed for higher throughput, reproducible SEC.

**Figure supplement 2.** Comparison of SEC resins by Western blotting.

**Figure supplement 3.** Effect of plasma sample volume on SEC.

**Figure supplement 4.** Comparison of EV recovery and albumin contamination across all tested methods in plasma.

**Figure supplement 5.** Coefficients of variation (CVs) across all tested methods in plasma.

it is necessary to combine several techniques sequentially, such as density gradient centrifugation (DGC) and SEC (*Karimi et al., 2018*; *Zhang et al., 2020*). However, techniques such as DGC are not scalable to many samples and therefore not amenable to biomarker studies. Thus, we focused on EV isolation methods that are amenable to biomarker studies. After finding that commercial SEC columns compare favorably to ultracentrifugation and ExoQuick precipitation, we compared several resins and column volumes to further improve EV isolation by custom SEC columns. SEC requires no instrumentation and allows one to run many columns in parallel. The throughput can be further increased by using SEC stands (such as the one we designed) and preparing columns ahead of time. Although we used freshly prepared columns in this study, we found comparable performance to columns stored at 4°C for 1 week.

Our investigation of SEC parameters led us to improved methods for EV isolation; in particular, we found that Sepharose CL-6B, which is seldom used for EV isolation, yields considerably higher levels of EVs than either Sepharose CL-2B, the most commonly used resin (*Monguió-Tortajada et al., 2019*), or Sepharose CL-4B. We attribute this result to Sepharose CL-6B beads having a smaller average pore size (reported [*Hagel et al., 1996*] to be 24 nm vs. 42 nm for CL-4B and 75 nm for CL-2B), leading to a lower probability that EVs will enter the beads. As there is a tradeoff between EV yield and albumin contamination, we envision different SEC columns will be suited for different applications. Using a 10-ml Sepharose CL-6B column for EV isolation from plasma or CSF is the best choice for downstream applications where maximum EV yield is needed and where some free protein contamination is not detrimental—for example, analyzing rare EV cargo or when further purification of EVs will be performed (such as immuno-isolation). On the other hand, if isolating EVs from plasma where minimal free protein contamination is desired (e.g., in EV protein analysis by Western blot), a larger 20 ml column with Sepharose CL-4B would yield better results. For CSF, which has much less protein than plasma, 10 ml columns are preferable to 20 ml ones.

By developing a Simoa assay to measure albumin (the most abundant protein in plasma and main contaminant when isolating EVs), we were able to assess the purity of EV preparations with respect to unwanted co-purification of free proteins. Our methods could be expanded to assess other contaminants that are less abundant than albumin but may, nonetheless, be problematic for some applications, such as lipoproteins. Adding a Simoa assay for ApoB100 (or other protein components of lipoproteins) would allow for the assessment of both lipoprotein and free protein contamination in EV isolation methods. Although lipoproteins are difficult to separate from EVs due to their overlapping size profile (*Simonsen, 2017*), a recent study demonstrated that a chromatography column combining a cation-exchange resin layer with an SEC resin layer allows for efficient lipoprotein depletion using 'dual mode chromatography' (*Van Deun et al., 2020*). Simoa could be used to evaluate and help improve such techniques in the future.

The general experimental framework presented here could be easily applied to evaluate new EV isolation methods in plasma, CSF, or other biological fluids, such as urine or saliva. While we limited our study to human biofluids, similar methods could also be applied to compare EV isolation methods from cell culture media. As sensitivity of EV detection and specificity in differentiating EVs from contaminants are obstacles in all EV studies, we envision that ultrasensitive protein detection with Simoa will be broadly applicable to assessing EV isolation methods for both the study of EV biology and development of EV diagnostics.

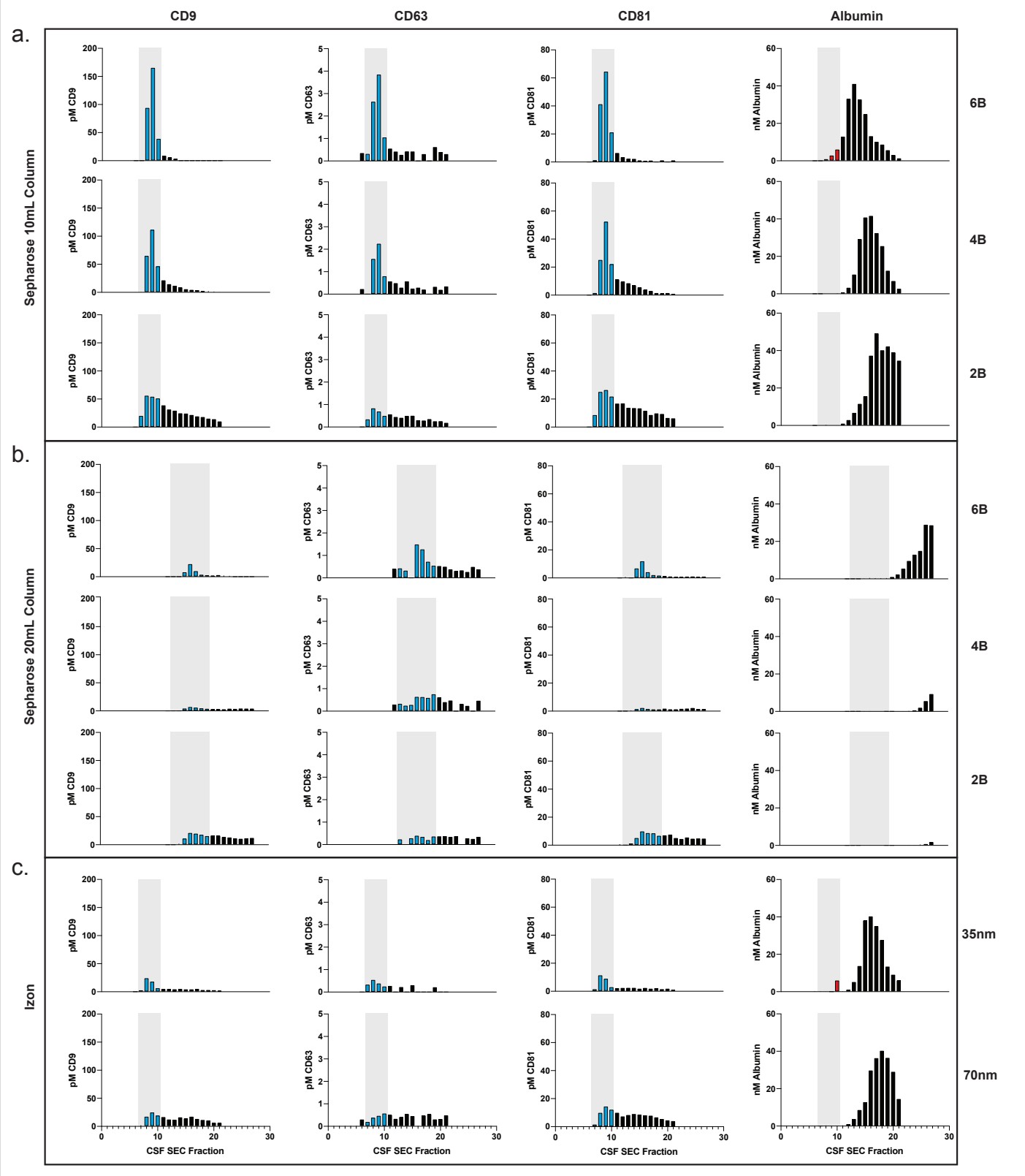

**Figure 4.** Comparison of SEC methods for EV isolation in CSF. (**a**) Levels of tetraspanins and albumin in CSF after fractionation with 10 ml custom columns filled with Sepharose CL-6B (top), Sepharose CL-4B (middle), and Sepharose CL-2B (bottom). (**b**) Levels of tetraspanins and albumin in CSF after fractionation with Izon qEVoriginal 35 nm column (top) and Izon qEVoriginal 70 nm column (bottom). (**c**) Levels of tetraspanins and albumin in CSF after fractionation with 20 ml custom columns; Sepharose CL-6B (top), Sepharose CL-4B (middle), and Sepharose CL-2B (bottom). CSF, cerebrospinal

*Figure 4 continued on next page*

*Figure 4 continued*

fluid; EV, extracellular vesicle; SEC, size exclusion chromatography.

The online version of this article includes the following source data and figure supplement(s) for figure 4:

**Source data 1.** CSF SEC optimization.

**Figure supplement 1.** Effect of CSF sample volume on SEC.

**Figure supplement 2.** Comparison of EV recovery and albumin contamination across all tested methods in CSF.

**Figure supplement 3.** Coefficients of variation (CVs) across all tested methods in CSF.

# Methods

## Key resources table

| Reagent type (species) or resource | Designation | Source or reference | Identifiers | Additional information |
|---|---|---|---|---|
| Biological sample (human) | Plasma | BioIVT | Cat #HUMANPLK2PNN | Pooled gender, K2EDTA |
| Biological sample (human) | Cerebrospinal fluid | BioIVT | Cat# HMNCSFR-NODXR | Pooled gender, no diagnosis remnant |
| Antibody | Anti-CD9 (Mouse monoclonal) | MilliporeSigma | Cat# CBL162 RRID:AB_2075914 | WB (1:1000) |
| Antibody | Anti-CD9 (Rabbit monoclonal) | Abcam | Cat# ab195422 RRID:AB_2893477 | Simoa capture |
| Antibody | Anti-CD9 (Mouse monoclonal) | Abcam | Cat# ab58989 RRID:AB_940926 | Simoa detector |
| Antibody | Anti-CD63 (Mouse monoclonal) | BD | Cat# 556019 RRID:AB_396297 | Simoa detector; WB (1:1000) |
| Antibody | Anti-CD63 (Mouse monoclonal) | R&D Systems | Cat# MAB5048 RRID:AB_2275726 | Simoa capture |
| Antibody | Anti-CD81 (Mouse monoclonal) | Thermo Fisher Scientific | Cat# 10630D RRID:AB_2532984 | WB (1:666) |
| Antibody | Anti-CD81 (Mouse monoclonal) | Abcam | Cat# ab79559 RRID:AB_1603682 | Simoa capture |
| Antibody | Anti-CD81 (Mouse monoclonal) | BioLegend | Cat# 349502 RRID:AB_10643417 | Simoa detector |
| Commercial assay or kit | Human Serum Albumin DuoSet ELISA | R&D Systems | Cat# DY1455 | Simoa capture and detector |
| Peptide, recombinant protein | CD9 | Abcam | Cat# ab152262 | |
| Peptide, recombinant protein | CD63 | Origene | Cat# TP301733 | |
| Peptide, recombinant protein | CD81 | Origene | Cat# TP317508 | |
| Peptide, recombinant protein | Albumin | Abcam | Cat# ab201876 | |
| Commercial assay or kit | ExoQuick exosome precipitation solution | SBI | Cat# EXOQ5A-1 | |
| Commercial assay or kit | ExoQuick ULTRA EV isolation kit for plasma and serum | SBI | Cat# EQULTRA-20A-1 | |
| Commercial assay or kit | qEVoriginal 70 nm | Izon | Cat# SP1 | |
| Commercial assay or kit | qEVoriginal 35 nm | Izon | Cat# SP5 | |
| Other | Sepharose CL-2B | Cytiva | Cat# 17014001 | |
| Other | Sepharose CL-4B | Cytiva | Cat# 17015001 | |
| Other | Sepharose CL-6B | Cytiva | Cat# 17016001 | |

## Human sample handling

Pre-aliquoted pooled human plasma (collected in K2 EDTA tubes) and CSF samples were ordered from BioIVT. The same pools were used for all main figures throughout the paper in order to ensure comparable analysis of methods. For all EV isolation technique comparisons, one 0.5-ml sample was used for each isolation method. Plasma or CSF was thawed at room temperature. After sample thawing, 100×

Protease/Phosphatase Inhibitor Cocktail (Cell Signaling Technology) was added to 1×. The sample was then centrifuged at 2000×*g* for 10 min. The supernatant was subsequently centrifuged through a 0.45 µm Corning Costar SPIN-X centrifuge tube filter (Sigma-Aldrich) at 2000×*g* for 10 min to get rid of any remaining cells or cell debris.

## Simoa assays

Simoa assays were developed and performed as previously described (**Norman et al., 2021**). A detailed protocol is available: https://www.protocols.io/view/simoa-extracellular-vesicle-assays-bm89k9z6. Capture antibodies were coupled to Carboxylated Paramagnetic Beads from the Simoa Homebrew Assay Development Kit (Quanterix) using EDC chemistry (Thermo Fisher Scientific). Detection antibodies were conjugated to biotin using EZ-Link NHS-PEG4 Biotin (Thermo Fisher Scientific). For CD9, ab195422 (Abcam) was used as capture antibody and ab58989 (Abcam) was used as detector antibody. For CD63, MAB5048 (R&D Systems) was used as capture antibody and 556019 (BD) was used as detector antibody. For CD81, ab79559 (Abcam) was used as capture antibody and 349502 (BioLegend) was used as detector antibody. For albumin, DY1455 (R&D Systems) was used as both capture and detector antibody. The following recombinant proteins were used for CD9, CD63, CD81, and albumin: ab152262 (Abcam), TP301733 (Origene), TP317508 (Origene), and ab201876 (Abcam). On-board dilution was performed with 4× dilution for each of the tetraspanins, while manual 20× dilution was used for albumin. All samples were raised to 160 µl per replicate in sample diluent. For tetraspanin assays, samples were incubated with immunocapture beads (25 µl) and biotinylated detection antibody (20 µl) for 35 min. Next, six washes were performed, and the beads were resuspended in 100 µl of Streptavidin labeled β-galactosidase (Quanterix) and incubated for 5 min. All bead washes were performed with Wash Buffer 1 (Quanterix). After incubation, an additional six washes were performed, and the beads were resuspended in 25 µl Resorufin β-D-galactopyranoside (Quanterix) before being loaded into the microwell array on the Quanterix HD-X instrument. For the albumin assay, samples were incubated first with immunocapture beads (25 µl) for 15 min and then washed six times. Subsequently, 100 µl detection antibody was incubated with the beads for 5 min. Next, six washes were performed, and the beads were resuspended in 100 µl of Streptavidin labeled β-galactosidase (Quanterix) for a final 5-min incubation. After an additional six washes, the beads were resuspended in 25 µl Resorufin β-D-galactopyranoside (Quanterix) and then loaded into the microwell array on the Quanterix HD-X instrument.

## Construction of SEC stand

The custom SEC rack was constructed from a total of 22 pieces using CNC milling tools. The rack is made of an aluminum frame (silver, Multipurpose 6061 Aluminum, McMaster-Carr) consisting of eight pieces, 4 sliding plates made from acetal (black, Wear-Resistant Easy-to-Machine Delrin Acetal Resin, McMaster-Carr), and 10 sliding plate grips made from UHMW Polyethylene (white, Slippery UHMW Polyethylene, McMaster-Carr). The rack frame is held together using 20 ¾" screws (McMaster-Carr, 92210 A113), 20 ½" screws (McMaster-Carr, 92210 A110), 10 0.375" Dowel pins (McMaster-Carr, 90145 A470), and 10 0.5625" Dowel pins (McMaster-Carr, 90145 A483), and includes 20 spring plungers (McMaster-Carr, 84895 A710) that allow the sliding plates to 'click' once aligned with the chromatography columns. Details for constructing the rack and SolidWorks files are included in the Supplementary materials.

## Preparation of custom SEC columns

**Table 1.** Recommendations for SEC columns for EV isolation from plasma and CSF.

|  | High yield | High purity |
| --- | --- | --- |
| Plasma | Sepharose CL-6B 10 ml column fractions 7–10 | Sepharose CL-4B 20 ml column fractions 14–17 |
| CSF | Sepharose CL-6B 10 ml column fractions 7–10 | Sepharose CL-4B 10 ml column fractions 7–10 |

The resins Sepharose CL-2B, Sepharose CL-4B, and Sepharose CL-6B (all from GE Healthcare/Cytiva) were washed in phosphate-buffered saline (PBS). The volume of resin was washed with an equal volume of PBS in a glass container and then placed at 4°C in order to let the resin settle completely (several hours or overnight). The PBS was then poured off, and an equal volume of PBS was again added two more times for a total of three washes. Columns were prepared fresh on

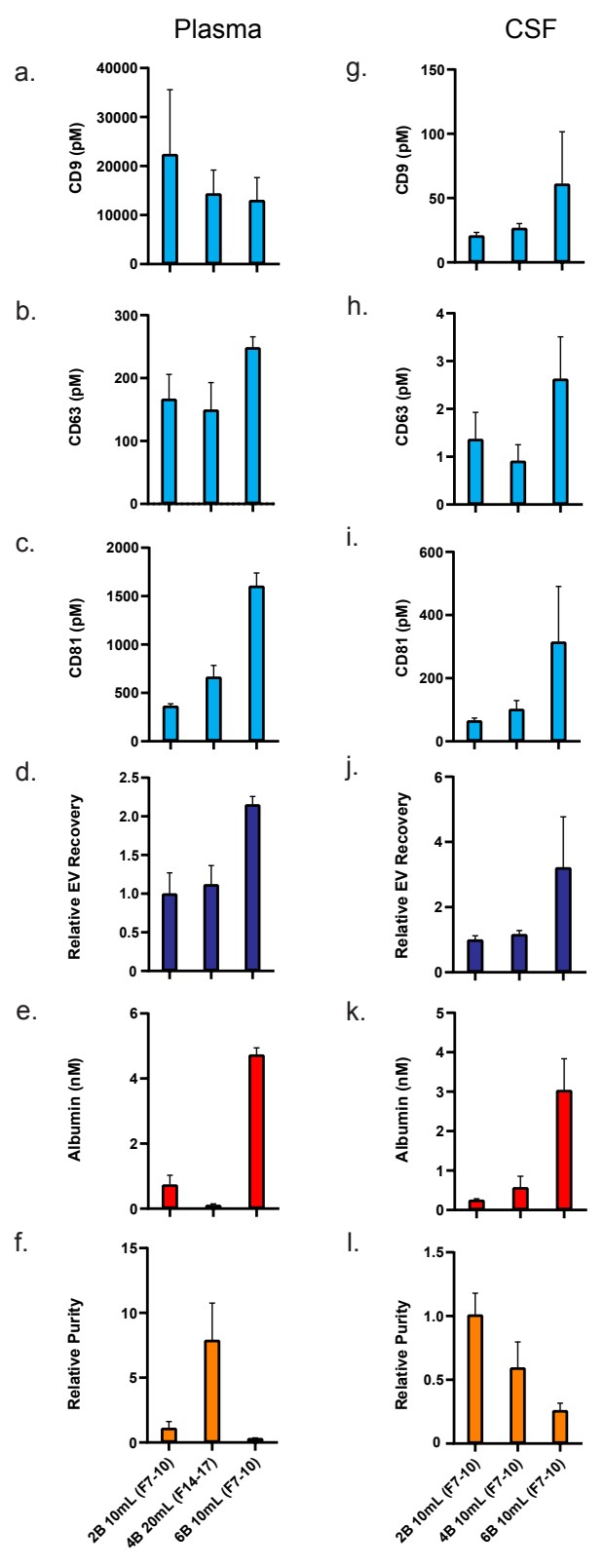

**Figure 5.** Comparison of top custom SEC methods in plasma and CSF. Error bars represent the standard deviations from four replicates of each column. (**a–c**). Individual tetraspanin yields using different isolation methods from plasma. (**d**) Relative EV recoveries from plasma were calculated by first normalizing individual tetraspanin values (in pM) in each technique to those of the Sepharose CL-2B 10 ml column (fractions 7–10) and

*Figure 5 continued on next page*

*Figure 5 continued*

then averaging the three tetraspanin ratios. (**e**) Albumin levels using different EV isolation methods from plasma. (**f**) EV purity for each method in plasma is calculated as the ratio of the sum of tetraspanin concentrations divided by albumin concentration. (**g–i**) Individual tetraspanin yield using different isolation methods from CSF. (**j**) Relative EV recoveries in CSF were calculated by first normalizing individual tetraspanin values (in pM) in each technique to those of Sepharose CL-2B 10 ml (fractions 7–10) and then averaging the three tetraspanin ratios. (**k**) Albumin levels using different EV isolation methods from CSF. (**l**) EV purity for each method in CSF is calculated as the ratio of the sum of tetraspanin concentrations divided by albumin concentration. CSF, cerebrospinal fluid; EV, extracellular vesicle; SEC, size exclusion chromatography.

The online version of this article includes the following source data for figure 5:

**Source data 1.** Top SEC methods in new batches of plasma and CSF.

the day of use. Washed resin was poured into an Econo-Pac Chromatography column (Bio-Rad) to bring the bed volume (the resin without liquid) to 10 or 20 ml. When the desired amount of resin filled the column and the liquid dripped through, the top frit was immediately placed at the top of the resin without compressing the resin. PBS was then added again before sample addition.

## Collection of size exclusion chromatography fractions

Once prepared, all columns were washed with at least 20 ml of PBS in the column. Immediately before sample addition, the column was allowed to fully drip out and, after last drop of PBS, sample (filtered plasma or CSF) was added to the column. As soon as sample was added, 0.5 ml fractions were collected in individual tubes. As soon as the plasma or CSF completely entered the column (below the frit), PBS was added to the top of column 1 ml at a time. Fraction numbers correspond to 0.5 ml increments collected as soon as sample is added. For Izon and 10 ml columns, fractions 6–21 were collected (since first few fractions correspond to void volume). For 20 ml columns, fractions 12–27 were collected (since void volume is larger for 20 ml columns than 10 ml columns). For **Figure 5**, only fractions 7–10 were collected.

## Ultracentrifugation

Samples of filtered 0.5 m plasma or CSF were added to 3.5 ml Open-Top Thickwall Polycarbonate ultracentrifuge tubes (Beckman Coulter), and PBS was added to fill tubes to the top. Samples were ultracentrifuged at 120,000×$g$ for 90 min at 4°C in an Optima XPN-80 ultracentrifuge (Beckman Coulter) using an SW55 Ti swinging-bucket rotor (Beckman Coulter). Afterward, all supernatant was aspirated. Pellets were resuspended in PBS for the 'Ultracentrifugation' condition. For the 'Ultracentrifugation with wash' condition, the ultracentrifuge tubes were filled to the top with PBS, and samples were ultracentrifuged again at 120,000×$g$ for 90 min. Supernatant was then aspirated, and pellets were resuspended in 500 μl PBS. For all ultracentrifugation samples, isolation was performed on 2 separate days and then resulting Simoa values were averaged.

## ExoQuick and ExoQuick ULTRA

Samples of plasma or CSF were mixed with ExoQuick Exosome Precipitation Solution (System Biosciences) or ExoQuick ULTRA EV Isolation Kit for Serum and Plasma (System Biosciences), and protocols were performed according to the manufacturer's instructions. For ExoQuick, 0.5 ml of plasma or CSF was mixed with 126 μl of ExoQuick and incubated at 4°C for 30 min, followed by centrifugation at 1500×$g$ for 30 min. Supernatant was removed, and samples were centrifuged at 1500×$g$ for an additional 5 min. Residual supernatant was removed, and pellets were resuspended in 500 μl PBS. For ExoQuick ULTRA, 250 μl of plasma or CSF was used in accordance with instructions, and Simoa values were corrected by multiplying by 2 to match the 0.5 ml volume used for other samples. For each sample, 500 μl of EVs was eluted per column. For all precipitations, isolation was performed on 2 separate days and then resulting Simoa values were averaged.

## Western blotting

Western blotting for tetraspanins was performed as previously described (**Kowal et al., 2017**), with minor modifications. 4× LDS was added to samples and samples were heated at 70°C for 10 min. Samples were run at 150 V for 1 hr on 4–12% Bolt Bis-Tris protein gels (Thermo Fisher Scientific)

and transferred using iBlot two nitrocellulose mini transfer stack (Thermo Fisher Scientific) at 20 V for 3 min. Blocking buffer was made by dissolving milk powder (to 5% w/v) in PBS-T (PBS with 0.1% Tween). Nitrocellulose membranes were blocked on a shaker for 30 min at 4°C, and then incubated with primary antibody overnight. The following antibodies and dilutions were used: 1:1000 BD (H5C6) for CD63, 1:1000 Millipore Clone MM2/57 (CBL162) for CD9 and 1:666 Thermo Fisher Scientific (M38) for CD81. Membranes were washed three times with PBS-T and incubated with 1:2000 human cross-adsorbed, anti-mouse HRP conjugated secondary antibody (Rockland) in blocking buffer for 2 hr. Membranes were washed three times with PBS-T and WesternBright ECL-spray HRP substrate (Advansta) was added. Images were acquired with a Sapphire Biomolecular Analyzer (Azure Biosystems).

## Acknowledgements

The authors acknowledge David Kalish for help designing and making the SEC stand and Emma Kowal for help with illustrations and comments on the manuscript. Funding for this study was provided by the Chan Zuckerberg Initiative (CZI) Neurodegeneration Challenge Network (NDCN) and Good Ventures Foundation. Schematics were created with BioRender.com

## Additional information

### Competing interests

Dmitry Ter-Ovanesyan, Maia Norman: The authors have filed intellectual property related to methods for isolating extracellular vesicles. George M Church: GMC commercial interests: http://arep.med.harvard.edu/gmc/tech.html. David R Walt: DRW has a financial interest in Quanterix Corporation, a company that develops an ultra-sensitive digital immunoassay platform. He is an inventor of the Simoa technology, a founder of the company and also serves on its Board of Directors. Dr. Walt's interests were reviewed and are managed by BWH. The authors have filed a provisional patent (WO2021163416A1) on methods for EV isolationmeasuring and purifying EVs. The other authors declare that no competing interests exist.

### Funding

| Funder | Grant reference number | Author |
|---|---|---|
| Chan Zuckerberg Initiative | NDCN Collaborative Science Award | Dmitry Ter-Ovanesyan<br>Maia Norman<br>Roey Lazarovits<br>Wendy Trieu<br>Ju-Hyun Lee<br>George Church<br>David R Walt |
| Open Philanthropy Project | | Dmitry Ter-Ovanesyan<br>Maia Norman<br>Roey Lazarovits<br>Wendy Trieu<br>Ju-Hyun Lee<br>David R Walt |

The funders had no role in study design, data collection and interpretation, or the decision to submit the work for publication.

### Author contributions

Dmitry Ter-Ovanesyan, Conceptualization, Investigation, Methodology, Validation, Writing – original draft, Writing – review and editing; Maia Norman, Conceptualization, Investigation, Methodology, Visualization, Writing – original draft, Writing – review and editing; Roey Lazarovits, Investigation, Methodology, Visualization, Writing – review and editing; Wendy Trieu, Ju-Hyun Lee, Investigation, Methodology; George M Church, Funding acquisition, Resources, Supervision; David R Walt, Funding acquisition, Resources, Supervision, Writing – review and editing

**Author ORCIDs**

Dmitry Ter-Ovanesyan http://orcid.org/0000-0002-1134-0073

Ju-Hyun Lee http://orcid.org/0000-0001-6728-2071

David R Walt http://orcid.org/0000-0002-5524-7348

**Decision letter and Author response**

Decision letter https://doi.org/10.7554/eLife.70725.sa1

Author response https://doi.org/10.7554/eLife.70725.sa2

## Additional files

### Supplementary files

- Transparent reporting form
- Source data 1. All data combined.

### Data availability

All data generated or analyzed during this study are included in the manuscript and supporting files.

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
