## [Decision Letter]

**Decision letter after peer review:**

Thank you for submitting your article "Framework for Rapid Comparison of Extracellular Vesicle Isolation Methods" for consideration by *eLife*. Your article has been reviewed by 3 peer reviewers, and the evaluation has been overseen by Y M Dennis Lo as the Senior and Reviewing Editor. The following individuals involved in review of your submission have agreed to reveal their identity: Qing Zhou (Reviewer #1); Julie A Saugstad (Reviewer #3).

Essential revisions:

1) The most novel aspect of this work is the strategy for quantifying EV yield and purity after EV isolation. To allow the authors to claim the potential superiority of their method, parallel analysis with a number of other currently available techniques (e.g. NTA) are necessary. For example, it would be useful to test if NTA can detect EV yield change between different isolation methods, or whether western blotting could show the same protein level changes reported by Simoa. There is also insufficient data to conclude that 10 mL Sepharose CL^-^6B column is the best choice for EV isolation from plasma and CSF.

2) A number of sub-conclusions are not solid enough. For example, whether different centrifugation speeds and time may affect the claim that SEC outperforms ultracentrifugation with regard to EV yield. It is well known that longer and higher speed centrifugation will increase the yield of EVs. Moreover, whether 10 mL Sepharose CL^-^6B column would still be the best choice for EV isolation from plasma or CSF when extending contamination markers to lipoprotein markers. As the size of lipoproteins is much larger, the purity of 10 mL Sepharose CL^-^6B column may be more affected than 10 mL Sepharose CL^-^2B columns.

3) Assessment of contamination could have been bolstered by examination of one or more proteins, e.g. lipoproteins.

4) The rationale for choosing some of the methods for comparison are missing – why test with and without the wash step for UC? Why use exoquick? The error bars are missing from most figures, the reproducibility of a method should be an important factor for consideration as a good isolation method.

5) Figure 3 should be reformatted as a supplement.

6) Frozen-thawed samples, which are commonly avoided in studies on EV, were used in this work. Why were such samples used, and would such samples produce bias to the authors' data?

7) The pre-analytical collection of the plasma and CSF should be described more fully in the Materials and methods and a couple of sentences added to the results. The pre-analytical steps are important for understanding the downstream results. While we note that the authors had purchased the samples from a company, we would like to know more details on the sample collection process. Information that would be useful includes: i) were EDTA collections tubes used? If not, state what type of tubes (Streck, etc) ii) was the plasma spun once to separate from buffy coat, or was it spun twice to also reduce platelets? iii) were hemoglobin/hemolysis measurements taken for the pools?

8) Figure 1C should have a graphic of each method tested. The graphic for SEC is clear and easy to understand. There should also be a short graphic for the ultracentrifugation with the speed of spins and the time laid out in order. There should also be a graphic for the exoquick – though that one is a little less exciting as a picture, but should describe incubation times etc.

9) More details should be included in the results describing the assays chosen for isolation – qEV 35 vs 70 and ultraC with or without wash and the two exoquicks. Explanations should be included why each of these was examined and what potential differences they represented – for example between 35 and 70, etc.

10) In Figure 2, the emphasis should be on recovery, not contamination. Consider reordering the results at the bottom based on relative EV recovery rather than albumin concentration. So in this case, d and e and i and j would switch places and the order would go from left to right on EV recovery. It would be useful to add error bars for this figure. An important component of comparing these assays concerns their reproducibility.

11) In the results – the authors claim that SEC is superior, but exoquick performs well in CSF. The authors should provide a set of characteristics which would objectively demonstrate that SEC is indeed superior to exoquick. Why would Izon 35 or 70 work differently in CSF compared with plasma? What might be some of the contributing factors influencing the effects of pore sizes that might aid in choosing one of these – or for understanding the albumin contamination in CSF.

12) The construction of the SEC collection apparatus is more appropriately placed in the supplementary materials. Izon sells and automated fraction collector, a similar stand, though not as sophisticated. Indeed, almost every lab that has done SEC often fashions its own collection apparatus. Thus, while the apparatus is interesting and well-constructed, its details can be regarded as supplemental.

13) A more detailed description of the pore sizes for the Sepharose chosen is needed. The testing of the columns for different size Sepharose beads and different heights is informative. Figures 4 and 5, one would like to more directly compare the 10 and 20 mL columns and would prefer they were A and B while Izon was C. And if the authors could make a measure of relative yield for the EV recovery, as they did in Figure 2 – that would be a nice additional column.

14) It would be useful to readers for the authors to summarize the outcomes of the plasma and CSF studies presented herein as "Recommendations" or as a summary table indicating the best method for each.

15) In Methods, Simoa Assays, first paragraph: It would be helpful to the reader to include the name of the Tetraspanins and detector antibodies before the catalog # and company name.

16) In Methods, Simoa Assays, second paragraph: In the sentence "Next, six washes were performed" it would be helpful to readers to say what buffer was used for the washes, and if these were consistently used for all the following washes.

17) In Methods, Preparation of Custom SEC Columns: "Columns were prepared fresh on the day of use". Can the authors comment in the Discussion whether the columns need to be prepared fresh on the day of use, or if they can be stored in the refrigerator, and if so, for how many days before use?

18) In Images, Figures 4 and 5, and Supplementary Figure 1: Please include higher resolution images. In Images, Supplementary Figure 2 is difficult to interpret as presented. Suggest grouping Relative albumin concentration by column and fraction number (6b, fx 7-9 and 7-10; Izon 35, fx 7-9 and 7-10), or some way other way to visualize effects of each method vs. the albumin concentration.

19) The authors should upload the Raw data from the Simoa runs as a supplementary file. Also, please include the Group Allocation information in the manuscript.

*Reviewer #1 (Recommendations for the authors):*

1) Add comparisons between this method with other commonly used techniques such as NTA to test the consistency and different performance.

2) Other contamination markers need to be included to strengthen the outperformance of this method, such as lipoprotein markers.

3) Re-evaluate the purity of different isolation methods with different contamination markers. May add one sample with a quicker and longer ultracentrifuge protocol to test the potential influence.

4) As noticed, frozen-thaw samples are used in this manuscript, usually avoided in EV studies. Can authors explain why not using fresh materials?

*Reviewer #2 (Recommendations for the authors):*

Introduction

The use of CD9, CD63, CD81 is appropriate for characterization of general EVs. Some data should be generated or discussed regarding abundance on different cell types or cell lines. It may be that these tetraspanins are expressed with different amounts in different contexts or cell types – or even on EVs released through different biogenesis pathways – in addition to not being present on every EV. The use of sensitive quantitative assays is very useful for understanding the purity and contamination with each method, though there are many papers out there already that do this with different assays and assess more EV components and contaminants.

Results

1) The pre-analytical collection of the plasma and CSF should be described more fully in the Materials and methods and a couple of sentences added to the results. The pre-analytical steps are important for understanding the downstream results. While you purchased the samples from a company – you can still report on the methods of collection. Information that should be included: 1) were EDTA collections tubes used? If not state what type of tubes (Streck, etc) 2) was the plasma spun once to separate from buffy coat, or was it spun twice to also reduce platelets? 3) were hemoglobin/hemolysis measurements taken for the pools? If aliquots remain in the freezer – you can add this information to the methods section on the samples.

2) Figure 1C should have a graphic of each method tested. The graphic for SEC is clear and easy to understand. There should also be a short graphic for the ultracentrifugation with the speed of spins and the time laid out in order. There should also be a graphic for the exoquick – though that one is a little less exciting as a picture, but should describe incubation times etc.

3) Spend a bit more time in the results describing the assays chosen for isolation – qEV 35 vs 70 and ultraC with or without wash and the two exoquicks. Provide some details on why each of these was examined and what potential differences they represent – for example between 35 and 70, etc.

4) In Figure 2, the emphasis should be on recovery, not contamination. Consider reordering the results at the bottom based on relative EV recovery rather than albumin concentration. So in this case, d and e and I and J would switch places and the order would go from left to right on EV recovery

5) Are there error bars for this? Can they be added? An important component of comparing these assays is also their reproducibility.

6) In the results – you claim SEC is superior, but exoquick performs well in CSF. You should provide a set of characteristics declaring it to be superior and why exoquick gets excluded. Why would Izon 35 or 70 work differently in CSF compared with plasma – what do you think are the contributing factors about the pore sizes that might aide in choosing one of these – or for understanding the albumin contamination in CSF. You do begin to answer this in the next section, but a more significant discussion is warranted.

7) I admire the construction of the SEC collection apparatus, but this should be supplementary material and should not be a main figure or part of the results. Izon sells and automated fraction collector, a similar stand, though not as sophisticated. And almost every lab that has done SEC very frequently has fashioned their own collection apparatus. So while this is interesting and well-constructed, it is supplemental.

8) More fully describe the pore sizes for the Sepharose chosen. The testing of the columns for different size Sepharose beads and different heights is informative. Figure 4 and 5, I would like to more directly compare the 10 and 20 mL columns and would prefer they were A and B while Izon was C. And if you could make a measure of relative yield for the EV recovery, as you did in Figure 2 – that would be a nice additional column.

9) With Figure 3 moved to supplemental, you could have the two supplemental figures become main figures.

10) Measuring a lipoprotein in addition to albumin would have been very useful to see.

Discussion

1) Why did you choose to look at plasma and CSF?

2) A comparison of Simoa with any other conventional assay would have been useful – like using exoView and each of the tetraspanins just to show similar counts for plasma one time.

*Reviewer #3 (Recommendations for the authors):*

The authors discuss comparing ultracentrifugation (with or without wash step) and two commercially available kits (ExoQuick and ExoQuick ULTRA), but do not include any data for these experiments. Although they state that this is because SEC was superior, it would be useful to include the outcomes from these studies at least in Supplementary data.

It would be useful to readers to summarize the outcomes of the plasma and CSF studies presented herein as "Recommendations" or as a summary table indicating the best method for each.

In Methods, Simoa Assays, first paragraph: It would be helpful to the reader to include the name of the Tetraspanins and detector antibodies before the catalog # and company name.

In Methods, Simoa Assays, second paragraph: In the sentence "Next, six washes were performed" it would be helpful to readers to say what buffer was used for the washes, and if these were consistently used for all the following washes.

In Methods, Preparation of Custom SEC Columns: "Columns were prepared fresh on the day of use" Can you comment in the discussion whether the columns need to be prepared fresh on the day of use, or if they can be stored in the refrigerator, and if so, for how many days before use?

In Images, Figures 4 and 5, and Supplementary Figure 1: Please include higher resolution images

In Images, Supplementary Figure 2 is difficult to interpret as presented. Suggest grouping Relative albumin concentration by column and fraction number (6b, fx 7-9 and 7-10; Izon 35, fx 7-9 and 7-10), or some way other way to visualize effects of each method vs. the albumin concentration.

I encourage the authors to upload the Raw data from the Simoa runs as a supplementary file. Also, please include the Group Allocation information in the manuscript.

---

## [Author Response]

Essential revisions:1) The most novel aspect of this work is the strategy for quantifying EV yield and purity after EV isolation. To allow the authors to claim the potential superiority of their method, parallel analysis with a number of other currently available techniques (e.g. NTA) are necessary. For example, it would be useful to test if NTA can detect EV yield change between different isolation methods, or whether western blotting could show the same protein level changes reported by Simoa. There is also insufficient data to conclude that 10 mL Sepharose CL^-^6B column is the best choice for EV isolation from plasma and CSF.

Thank you to the reviewers for these comments. We certainly agree that different methods for comparing EV isolation methods would yield different results. Since there is no perfect technique for measuring EVs, there is also no true positive control to compare to. In particular, NTA has been widely recognized to be problematic for quantifying EVs due to its inability to discriminate between EVs and the similarly sized but more abundant non-EV particles such as lipoproteins or protein aggregates (1-3). For example, Welton et al. demonstrated that there is poor overlap between particles measured by NTA and tetraspanin levels after fractionating plasma by SEC, leading the authors to conclude that >70% of particles measured by NTA in plasma are not EVs (4). This was our motivation for using Simoa to measure tetraspanin levels, which overcomes this limitation of NTA and other particle counting methods. Of course, measuring specific proteins, which are not present on every single EV is a limitation of our approach, but since we are comparing EV isolation methods from the same starting sample of biofluid, Simoa enables an “apples to apples” comparison that is quantitative in a way that other techniques (such as NTA) are not. We have previously compared Simoa to ELISA and western blot (in Supplemental Figure 2) from the same sample (5). To show concordance between Simoa and western blot, we repeated the comparison of 10mL SEC columns with Sepharose CL^-^2B, CL^-^4B, and CL^-^6B and evaluated the EV fractions (F7-10) by western blotting for CD9, CD63, and CD81. Our western blot results confirm that using CL^-^6B leads to the highest levels of tetraspanins and that SEC resins with smaller pore sizes lead to more EVs. We have added a supplemental figure with these results. Whether or not Sepharose CL^-^6B is the best choice, as we state in the paper, depends on the goal. Using this resin yields more EVs, but also more albumin contamination.

2) A number of sub-conclusions are not solid enough. For example, whether different centrifugation speeds and time may affect the claim that SEC outperforms ultracentrifugation with regard to EV yield. It is well known that longer and higher speed centrifugation will increase the yield of EVs. Moreover, whether 10 mL Sepharose CL^-^6B column would still be the best choice for EV isolation from plasma or CSF when extending contamination markers to lipoprotein markers. As the size of lipoproteins is much larger, the purity of 10 mL Sepharose CL^-^6B column may be more affected than 10 mL Sepharose CL^-^2B columns.

Thank you to the reviewers for these comments. The purpose of our study was not to comprehensively compare all EV isolation methods, as there are many isolation methods (and many variations of each, such as different ultracentrifugation times). Our main goal was to propose a framework for rapidly comparing EV isolation methods using quantitative measurements. It has, indeed, been shown that increasing centrifugation time increases EV yield, but also free protein contamination (6). As we found SEC to be superior to ultracentrifugation in our initial survey of different isolation methods, and SEC is much higher throughput than ultracentrifugation, we decided to demonstrate the utility of our technique by applying it to the optimization of SEC. We certainly agree that albumin is not the only contaminant in EV preparations and there are other contaminants, such as lipoproteins. Additionally, separating EVs from lipoproteins is acknowledged to be challenging due to their overlapping size ranges (4, 7, 8). We agree that it is likely different SEC resins would have different effects on separation of lipoproteins and EVs. We chose albumin as it is the most abundant protein in plasma, and, thus, the main contaminant in EV preparations. As we mention in the Discussion section of our manuscript, for future studies, we plan to add other Simoa assays for ApoB100 or other protein components of lipoproteins and optimize the separation of EVs from both lipoproteins and free proteins.

3) Assessment of contamination could have been bolstered by examination of one or more proteins, e.g. lipoproteins.

As mentioned in #2, we agree with the reviewer’s comments that there is more than one type of contamination when separating EVs from plasma. Although we plan to develop Simoa assays for protein components of lipoproteins in future studies, this was beyond the scope of this study, which was to demonstrate the utility of Simoa to quantitatively compare EV isolation methods in terms of yield and free protein contamination. We chose albumin as it is the most abundant protein in plasma, and also a “model” contaminating, free protein. In subsequent studies, we hope to use Simoa assays to specifically optimize separation of EVs from lipoproteins. It would be interesting to build on the results of our study, for example, to see if we can incorporate Sepharose CL^-^6B into improved “dual-mode chromatography” columns, which have been previously shown to deplete lipoproteins when combined with a cation exchange resin (9). We plan to explore these questions in the future.

4) The rationale for choosing some of the methods for comparison are missing – why test with and without the wash step for UC? Why use exoquick? The error bars are missing from most figures, the reproducibility of a method should be an important factor for consideration as a good isolation method.

Thank you to the reviewers for these comments. We initially began by comparing the EV isolation methods that are most widely used in the context of biomarker studies for human biofluids. These commonly used methods are ultracentrifugation, precipitation (ExoQuick), and SEC (10). For each of these methods, we chose a common variation: ultracentrifugation is sometimes done with or without a wash step, ExoQuick has a new kit with an extra spin column step that the manufacturers claim gives higher purity (ExoQuick Ultra), and Izon offers two SEC columns (but it is unclear how the yield of EVs compares between the two). After performing a broad survey of these techniques to get a general sense of how they compare, we saw that the commercial SEC columns performed favorably, and we set out to further explore how the various parameters (resin type and volume) affect EV yield and purity using homemade SEC columns. All of these experiments were done using the same pooled and aliquoted batches of plasma and CSF. Using these samples allowed us to directly compare the Simoa measurements to each other. For the ultracentrifugation and precipitation-based methods (where one final sample is collected), we performed isolations on two different days and then performed two technical replicates on each isolation. Because there was a limit to how much pooled CSF we could obtain, and how many Simoa jobs we could reasonably run, we included SEC results from one column per condition, collecting all fractions with two technical replicates per measurement. Although we have performed many column comparisons with other batches of biofluids, we did not include those, as they could not be directly compared to the ones we included the manuscript. Instead, using a new batch of pooled CSF and plasma, we took the best conditions from the broad survey isolation methods, and redid the comparison of high purity and high yield EV isolation methods (shown in Figure 6). For that comparison, we did four replicates for each column, and two technical replicates of each condition, giving us enough data to include error bars. For figure 2, we did not feel that the low number of replicates warranted the addition of error bars. But to address variability, we have added additional supplementary figures: Figure 2—figure supplement 1 to show the variability of precipitation and ultracentrifugation performed on two different days and Figure 3—figure supplement 5 and Figure 4—figure supplement 3 to show the variability of technical replicates across all conditions tested in Figures 2-5.

5) Figure 3 should be reformatted as a supplement.

Thank you for the suggestion. We have moved Figure 3 to the supplement in the revised manuscript.

6) Frozen-thawed samples, which are commonly avoided in studies on EV, were used in this work. Why were such samples used, and would such samples produce bias to the authors' data?

We agree that pre-analytical variables such as freeze-thaw samples are very important to consider when thinking about biomarker studies. In this case, we wanted to evaluate samples that were as close as possible to samples used in clinical biomarker studies. Since such samples always undergo at least one freeze-thaw cycle, we decided to use pooled CSF and plasma that was shipped to us frozen to mimic clinical samples.

7) The pre-analytical collection of the plasma and CSF should be described more fully in the Materials and methods and a couple of sentences added to the results. The pre-analytical steps are important for understanding the downstream results. While we note that the authors had purchased the samples from a company, we would like to know more details on the sample collection process. Information that would be useful includes: i) were EDTA collections tubes used? If not, state what type of tubes (Streck, etc) ii) was the plasma spun once to separate from buffy coat, or was it spun twice to also reduce platelets? iii) were hemoglobin/hemolysis measurements taken for the pools?

Thank you for the suggestion. We have expanded the Materials and methods section to include information about the type of plasma samples that we ordered from BioIVT (EDTA collection tubes). The plasma was spun once to separate from buffy coat prior to shipment to us. However, upon thawing, as mentioned in the Materials and methods, we performed an additional spin at 2000g for 10 minutes and filtered plasma (or CSF) through a 0.45um filter. Hemoglobin/hemolysis measurements were not taken for the pools.

8) Figure 1C should have a graphic of each method tested. The graphic for SEC is clear and easy to understand. There should also be a short graphic for the ultracentrifugation with the speed of spins and the time laid out in order. There should also be a graphic for the exoquick – though that one is a little less exciting as a picture, but should describe incubation times etc.

Thank you for the suggestion. We have added a graphic to illustrate the details of each method tested as Figure 2a.

9) More details should be included in the results describing the assays chosen for isolation – qEV 35 vs 70 and ultraC with or without wash and the two exoquicks. Explanations should be included why each of these was examined and what potential differences they represented – for example between 35 and 70, etc.

Thank you for the suggestion. As mentioned in #4, we initially chose methods that were the most commonly used in context of EV biomarker studies. For each of these methods, we picked one common variation for each. For example, Izon offers two commercial columns and their website states that the 70nm columns is better for isolating large EVs relative to the 35nm column, but no additional information or data are provided. We omitted techniques such as density gradient centrifugation, for example, since this technique is not amenable to biomarker studies where large numbers of samples are used. We have added a few sentences in the Results section of our manuscript providing additional context for this.

10) In Figure 2, the emphasis should be on recovery, not contamination. Consider reordering the results at the bottom based on relative EV recovery rather than albumin concentration. So in this case, d and e and i and j would switch places and the order would go from left to right on EV recovery. It would be useful to add error bars for this figure. An important component of comparing these assays concerns their reproducibility.

Thank you to the reviewer for the suggestions. We reordered the conditions in Figure 2 based on the recovery. As we mentioned in #4, We agree that reproducibility is important and have added two supplemental figures to address this. We added Figure 2—figure supplement 1 to show experimental variability of the non-SEC methods performed on two independent days with the same batch of plasma or CSF. We also added supplementary figures (Figure 3—figure supplement 5 and Figure 4—figure supplement 3) to demonstrate the variability (CVs) of technical replicates (same sample measured twice by Simoa) for all measurements performed in Figures 2-5.

11) In the results – the authors claim that SEC is superior, but exoquick performs well in CSF. The authors should provide a set of characteristics which would objectively demonstrate that SEC is indeed superior to exoquick. Why would Izon 35 or 70 work differently in CSF compared with plasma? What might be some of the contributing factors influencing the effects of pore sizes that might aid in choosing one of these – or for understanding the albumin contamination in CSF.

We also found it interesting that some isolation methods showed a difference in relative performance in plasma compared CSF. Since plasma has a protein concentration that is approximately two orders of magnitude higher than that of CSF, we think this is the most likely reason for the difference. Our data indicating that Exoquick works relatively better in CSF compared to plasma could be due to CSF having much less albumin than plasma. When comparing all of the techniques we tested in the initial broad survey (including the custom SEC columns), as shown in Figure 4—figure supplement 2, we found that Sepharose CL^-^6B had higher relative EV yield than ExoQuick in CSF.

12) The construction of the SEC collection apparatus is more appropriately placed in the supplementary materials. Izon sells and automated fraction collector, a similar stand, though not as sophisticated. Indeed, almost every lab that has done SEC often fashions its own collection apparatus. Thus, while the apparatus is interesting and well-constructed, its details can be regarded as supplemental.

We have moved Figure 3 to the supplement in the revised manuscript as Figure 3—figure supplement 1.

13) A more detailed description of the pore sizes for the Sepharose chosen is needed. The testing of the columns for different size Sepharose beads and different heights is informative. Figures 4 and 5, one would like to more directly compare the 10 and 20 mL columns and would prefer they were A and B while Izon was C. And if the authors could make a measure of relative yield for the EV recovery, as they did in Figure 2 – that would be a nice additional column.

Thank you to the reviewer for these suggestions. We have included the sizes of the pores for the different resins in our Discussion section. We have also moved Izon and the 10mL column, as suggested, to more easily compare the 10mL and 20mL columns. For relative EV recovery for the SEC columns, that comparison is included in Figure 3—figure supplement 4 and Figure 4—figure supplement 2 comparing all of the different SEC methods to each other and the non-SEC methods.

14) It would be useful to readers for the authors to summarize the outcomes of the plasma and CSF studies presented herein as "Recommendations" or as a summary table indicating the best method for each.

We have included a table indicating our recommendations for “high purity” and “high yield” EV isolation from both plasma and CSF.

15) In Methods, Simoa Assays, first paragraph: It would be helpful to the reader to include the name of the Tetraspanins and detector antibodies before the catalog # and company name.

We have changed the manuscript to incorporate these changes.

16) In Methods, Simoa Assays, second paragraph: In the sentence "Next, six washes were performed" it would be helpful to readers to say what buffer was used for the washes, and if these were consistently used for all the following washes.

We clarified the wash buffer we used and state that the same buffer was used for all bead washes. We have also provided a link to a more detailed description of the methods: https://www.protocols.io/view/simoa-extracellular-vesicle-assays-bm89k9z6

17) In Methods, Preparation of Custom SEC Columns: "Columns were prepared fresh on the day of use". Can the authors comment in the Discussion whether the columns need to be prepared fresh on the day of use, or if they can be stored in the refrigerator, and if so, for how many days before use?

For this paper, we prepared all SEC columns fresh on the date of use, but we compared fresh columns to those stored for one week in the refrigerator and found that they gave similar results by Simoa. We add this comment to the Discussion.

18) In Images, Figures 4 and 5, and Supplementary Figure 1: Please include higher resolution images. In Images, Supplementary Figure 2 is difficult to interpret as presented. Suggest grouping Relative albumin concentration by column and fraction number (6b, fx 7-9 and 7-10; Izon 35, fx 7-9 and 7-10), or some way other way to visualize effects of each method vs. the albumin concentration.

In the revised manuscripts, we have replaced with high resolution images. We also added additional figures to Supplemental Figures S5 and S6 with all of the methods in the initial comparison ordered by relative albumin concentration in addition to by relative EV recovery.

19) The authors should upload the Raw data from the Simoa runs as a supplementary file. Also, please include the Group Allocation information in the manuscript.

We uploaded all raw data as a supplementary file. We now describe the Group Allocation information in the Materials and methods section and also in the Transparent Reporting Form.

References:

1. Coumans FAW, Brisson AR, Buzas EI, Dignat-George F, Drees EEE, El-Andaloussi S, et al. Methodological Guidelines to Study Extracellular Vesicles. Circulation research. 2017;120(10):1632-48.2. Hartjes TA, Mytnyk S, Jenster GW, van Steijn V, van Royen ME. Extracellular Vesicle Quantification and Characterization: Common Methods and Emerging Approaches. Bioengineering (Basel, Switzerland). 2019;6(1).3. Johnsen KB, Gudbergsson JM, Andresen TL, Simonsen JB. What is the blood concentration of extracellular vesicles? Implications for the use of extracellular vesicles as blood-borne biomarkers of cancer. Biochim Biophys Acta Rev Cancer. 2019;1871(1):109-16.4. Welton JL, Webber JP, Botos LA, Jones M, Clayton A. Ready-made chromatography columns for extracellular vesicle isolation from plasma. Journal of extracellular vesicles. 2015;4:27269.5. Norman M, Ter-Ovanesyan D, Trieu W, Lazarovits R, Kowal EJK, Lee JH, et al. L1CAM is not associated with extracellular vesicles in human cerebrospinal fluid or plasma. Nature methods. 2021;18(6):631-4.6. Cvjetkovic A, Lötvall J, Lässer C. The influence of rotor type and centrifugation time on the yield and purity of extracellular vesicles. Journal of extracellular vesicles. 2014;3.7. Simonsen JB. What Are We Looking At? Extracellular Vesicles, Lipoproteins, or Both? Circulation research. 2017;121(8):920-2.8. Sodar BW, Kittel A, Paloczi K, Vukman KV, Osteikoetxea X, Szabo-Taylor K, et al. Low-density lipoprotein mimics blood plasma-derived exosomes and microvesicles during isolation and detection. Scientific reports. 2016;6:24316.9. Van Deun J, Jo A, Li H, Lin HY, Weissleder R, Im H, et al. Integrated Dual-Mode Chromatography to Enrich Extracellular Vesicles from Plasma. Advanced biosystems. 2020:e1900310.10. Monguio-Tortajada M, Galvez-Monton C, Bayes-Genis A, Roura S, Borras FE. Extracellular vesicle isolation methods: rising impact of size-exclusion chromatography. Cellular and molecular life sciences : CMLS. 2019;76(12):2369-82.